# Achieving ultrahigh instantaneous power density of 10 MW/m² by leveraging the opposite-charge-enhanced transistor-like triboelectric nanogenerator (OCT-TENG)

Hao Wu [1,2], Steven Wang[2], Zuankai Wang [2✉] & Yunlong Zi [1✉]

Converting various types of ambient mechanical energy into electricity, triboelectric nano-generator (TENG) has attracted worldwide attention. Despite its ability to reach high open-circuit voltage up to thousands of volts, the power output of TENG is usually meager due to the high output impedance and low charge transfer. Here, leveraging the opposite-charge-enhancement effect and the transistor-like device design, we circumvent these limitations and develop a TENG that is capable of delivering instantaneous power density over 10 MW/m² at a low frequency of ~ 1 Hz, far beyond that of the previous reports. With such high-power output, 180 W commercial lamps can be lighted by a TENG device. A vehicle bulb containing LEDs rated 30 W is also wirelessly powered and able to illuminate objects further than 0.9 meters away. Our results not only set a record of the high-power output of TENG but also pave the avenues for using TENG to power the broad practical electrical appliances.

[1] Department of Mechanical and Automation Engineering, The Chinese University of Hong Kong, Hong Kong, China. [2] Department of Mechanical Engineering, City University of Hong Kong, Hong Kong, China. ✉email: zuanwang@cityu.edu.hk; ylzi@cuhk.edu.hk

Nowadays, our world faces unprecedented threats from the energy crisis, global warming, and environmental pollution due to the heavy reliance on fossil fuels[1,2]. For optimizing the world's energy consumption structure, attentions lie in the vast clean and renewable energies contained in the mechanical motions in the environment, such as water waves, wind, rain droplets, and biomechanical motions. Particularly, with the recent emergence of the Internet of Things, these omnipresent environmental mechanical energies may perfectly serve as the power sources for the numerous sensors distributed worldwide. Being able to harvest these ambient mechanical energies[3–7], the triboelectric nanogenerator (TENG) based on the coupling of triboelectrification and electrostatic induction has immediately attracted widespread interest with its favored advantages of easy fabrication, light-weight, magnet-free, and low-frequency mechanical energy harvesting capability[8–10], since it was proposed in 2012[8]. Consisting of only several thin films, TENG can convert various mechanical energy sources into electricity and easily generate an open-circuit voltage of kV level[11,12].

However, TENG suffers from two fundamental limitations: the low charge transfer[13–16] (~100 $\mu C/m^2$) and the high output impedance[13,17] (in the order of M$\Omega$), which result in low output power. The charge output of the TENGs can be increased by enhancing tribo-material's surface charge density[18–27], but it often needs extra material modification processes[26,27], external charge excitation modules[24,25], or strict environmental conditions[21,28]. On the other hand, the impedance can be reduced by employing power management (PM) circuits[13,29,30]. For example, with the most recent state-of-the-art PM circuit design, instantaneous power density reached 11.13 kW/m$^2$, and 10 W commercial lamps can be powered with an effective area of 100 cm$^2$ [29]. Yet, besides the drawbacks of the bulkiness of the PM circuits with several transistors, capacitors, and other elements, these electronic components also consume a considerable share of the output power and thus hindered the power delivered on the load devices. Therefore, enhancing the total charge transfer and diminishing the output impedance are the keys for TENG to serve as a high-performance power supply.

In this work, we develop an opposite-charge-enhanced transistor-like TENG (OCT-TENG) that is capable of delivering ultrahigh instantaneous power density over 10 MW/m$^2$. The attainment of both opposite charges (positive and negative) via coplanar opposite tribo-surfaces delivers a largely enhanced charge transfer. Meanwhile, the design of the transistor-like structure imparts a close-to-zero output impedance, thereby enabling the amplified charges to be efficiently released through the external load. Using an OCT-TENG with an effective area of 25 cm$^2$, commercial lamps rated up to 180 W can be lighted up. A vehicle bulb containing high-power LEDs (30 W) can also be wirelessly powered and brightly illuminate objects further than 0.9 m away. Surpassing the previous reports using mW-level LEDs for demonstrating the TENG's performance, our results have paved the avenues for using TENG to power the broad practical applications.

## Results

### High-performance of the OCT-TENG.
As shown in Fig. 1a and Supplementary Fig. 1, the OCT-TENG consists of a stator substrate and a slider. The stator contains coplanar tribo-surfaces with opposite-charge polarization (fluorinated ethylene propylene, refer as FEP, and polycarbonates, refer as PC, both with the thickness of 100 $\mu$m) and four electrodes ($E_1$, $E_2$, $E_L$, and $E_R$). The $E_L$ and $E_R$ are two floating electrodes placed at the left and right sides of the stator substrate. The $E_1$ and $E_2$ are two sheet elec-

trodes (Cu) placed beneath the FEP and PC films. The slider contains a sheet electrode $E_3$ beneath a thin FEP film (10 $\mu$m). The whole device is analogous to a complementary transistor pair of two structurally identical transistor-like parts with opposite tribo-surfaces. $E_1$ and $E_2$ can be regarded as the "source" (S) of each "transistor", and $E_3$ as the dynamic "drain" (D). The $E_L$ and $E_R$ are the "gate" (G). When $E_3$ touches the $E_L$ or $E_R$, the "gate" of the transistor is switched "ON", and accordingly, the charges can flow between the "source" and the "drain". For comparison, we also fabricated a conventional sliding mode TENG (the "control-TENG" shown in Fig. 1b and Supplementary Fig. 2) and a TENG with a unidirectional switch (TENG-UDS)[31] (Supplementary Fig. 3).

The measured OCT-TENG's peak output current across the load resistor (R = 1 M$\Omega$) is ~2.7 mA, around 300 times higher than that of the control-TENG using the same slider, as shown in Fig. 1c and Supplementary Fig. 4. More interestingly, with the total charge transfer of around 2.5 times of the control-TENG, our OCT-TENG can deliver several orders of magnitude higher power, as shown in Fig. 1d and Supplementary Figs. 5–6. Ultrahigh current of 37 A can be achieved with effective area of 25 cm$^2$ on a load resistance of 22 $\Omega$, corresponding to instantaneous power density of ~30 kW (12 MW/m$^2$), as shown in Fig. 1d and Supplementary Fig. 7. Compared to the TENG-UDS, the outputs of our OCT-TENG are also much higher. The pulsed power and average power outputs of the TENG-UDS are only ~ 4% and 1.5% of that of the OCT-TENG, respectively (Supplementary Figs. 4–6). With load resistance ranging from 22 $\Omega$ to 120 $\Omega$, the instantaneous power density of our OCT-TENG keeps above 10 MW/m$^2$, being far beyond that in the literature[13,19,28,29,32,33] (Fig. 1e). Moreover, with a wide range of load resistance from 22 $\Omega$ to 10 M$\Omega$, the average power densities of our OCT-TENG are all above 450 mW·m$^{-2}$·Hz$^{-1}$, and the maximum value reaches 790 mW·m$^{-2}$·Hz$^{-1}$, which is also higher than that of the recent record of 110 mW·m$^{-2}$·Hz$^{-1}$ [29](see below and Supplementary Fig. 8). Using our OCT-TENG, 180 W commercial lamps can be lighted up, as shown in Fig. 1f.

### The working mechanism of the OCT-TENG.
To clearly explain the significant enhancement of power output of our OCT-TENG over the conventional TENG devices, we divided the operation and power generation process of the OCT-TENG into four stages, as shown in Fig. 2a. At stage 1, the slider is on top of the FEP surface, and the electrode $E_3$ contacts the left floating electrode $E_L$, corresponding to the "ON" state of the left "transistor". At stage 2, the slider moves towards the PC side (before contacting the $E_R$). At this stage, the "transistors" are at the "OFF" state. At stage 3, the $E_3$ contacts the $E_R$, corresponding to the "ON" state of the "transistor" on the right side. At stage 4, the slider moves towards the PC side before $E_3$ touching $E_L$; the "transistors" are at the "OFF" state. At each stage, electrons transfer between the corresponding electrodes and balance the established potential difference, as shown in Fig. 2a. The equivalent circuits are shown in Supplementary Fig. 9. At the "ON" state (stages 1 and 3), the charges immediately transfer between the corresponding "source" ($E_1$ or $E_2$) to the "drain" ($E_3$). At the "OFF" state (stages 2 and 4), the charges transfer between the two "sources" ($E_1$ and $E_2$) along with the slider moving. Benefiting from the smart circuit design, at all these four stages, electric charges transfer between the point "A" and point "B" in the circuit (labeled in Fig. 2a) and deliver electricity to the external load.

We calculate the amount of charge transfer based on the proposed mechanism (Note 1 in the Supplementary Information), and the charge transfer at these four stages can be expressed

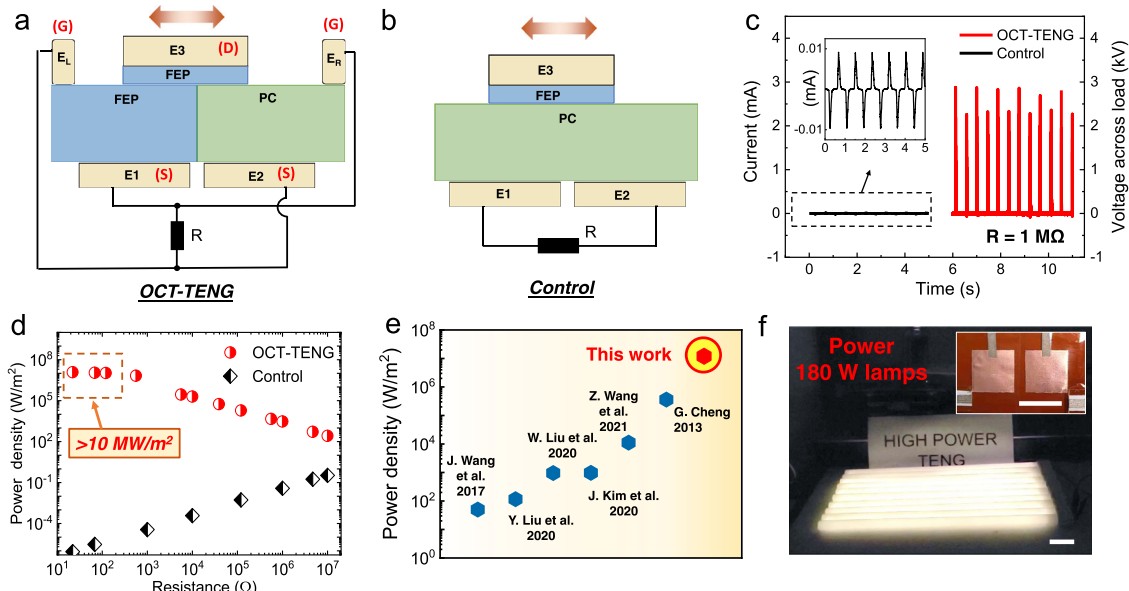

**Fig. 1 Design and performance of the OCT-TENG.** Schematic of the cross-section of **a** opposite-charge-enhanced transistor-like triboelectric nanogenerator (OCT-TENG) and **b** the control device of conventional sliding mode TENG. **c** Comparison of current output between OCT-TENG and the control device. (load resistance $R$ of 1 MΩ) **d** Comparison of pulsed power depending on load resistance between OCT-TENG and the control device. **e** Comparison of the instantaneous power density obtained in this work with other reports[13,19,28,29,32,33]. **f**, the photograph of 180 W lamps being powered by an OCT-TENG. Insert shows the photograph of the OCT-TENG. Scale bars in **f** and insert are 5 cm.

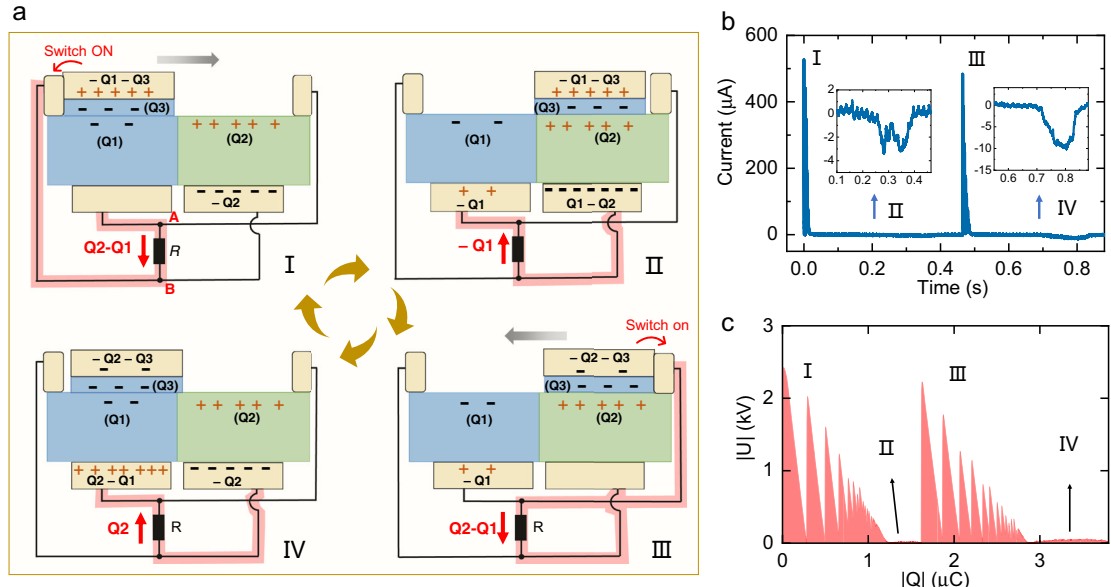

**Fig. 2 Working principle of the OCT-TENG. a** Schematic diagram of the working principle of the OCT-TENG. The charge transfer path in each stage is labeled in red. **b** The generated current and **c** U–Q plot of the OCT-TENG during one cycle of operation (load resistance of 4.7 MΩ).

as:

$$Q_{s1} = Q_{s3} = (Q_3 + Q_2)\frac{\varepsilon_3 d_2}{\varepsilon_3 d_2 + \varepsilon_2 d_3} - (Q_3 + Q_1)\frac{\varepsilon_3 d_1}{\varepsilon_3 d_1 + \varepsilon_1 d_3} \quad (1)$$

$$Q_{s2} = \frac{\varepsilon_3 d_1 Q_1 - \varepsilon_1 d_3 Q_3}{\varepsilon_3 d_1 + \varepsilon_1 d_3} \quad (2)$$

$$Q_{s4} = \frac{\varepsilon_2 d_3 Q_3 - \varepsilon_3 d_2 Q_2}{\varepsilon_3 d_2 + \varepsilon_2 d_3} \quad (3)$$

where $Q_1, Q_2$ and $Q_3$ are the amounts of charges on the FEP

surface of the stator, the PC surface, and the FEP surface of the slider, respectively. $\varepsilon_1, \varepsilon_2$, and $\varepsilon_3$ are the correspondence dielectric constants of the tribo-materials. $d_1, d_2$, and $d_3$ are the correspondence thickness of these tribo-materials. According to Eqs. (1)–(3), the maximum charge transfer happens when $d_1, d_2 \gg d_3$, which is the condition used in this work. Considering $Q_1 < 0$, and $Q_2 > 0$ due to polarization of the FEP and PC, Eqs. (1)–(3) can be simplified as $Q_{s1} = Q_{s3} \approx Q_2 - Q_1$, $Q_{s2} \approx Q_1$, and $Q_{s4} \approx -Q_2$, respectively. Benefiting from the opposite-charge-enhancement effect, at the "ON" state (stages 1 or 3), large amount of charges $(Q_2 - Q_1)$ flows in the same current direction from "A" to "B". Together with the charge transfer at the "OFF"

state (stages 2 and 4) of $|Q_1|$ $(=-Q_1)$ and $|Q_2|$ $(=Q_2)$, the total charge transfer in our OCT-TENG per cycle makes $3(|Q_2|+|Q_1|)$, and that is much higher than that of control-TENG and TENG-UDS (Supplementary Figs. 5 and 6).

The experimental results well match our proposed mechanism and the model. Benefitting from the opposite-charge-enhancement effect and the transistor-like structure, at the "ON" state (stages 1 and 3), large amount of charges $(Q_2 - Q_1)$ quickly transfer from the "source" to the "drain", leading to sharp and high current peaks, as shown in Fig. 2b and Supplementary Fig. 10. Such high current results in high energy generation, as shown as the area under the jagged curve in the U–Q plot shown in Fig. 2c. This process can be analogous to the droplet-based electricity generator (DEG)[3,4,23,34,35] when the drop touches the top electrode, as well as the conventional TENG-UDS[31] at the moment when the electrodes on the slider contact the stationary electrodes. Surpassing the DEG and the TENG-UDS, the generated power of our OCT-TENG is much higher due to the opposite-charge-enhancement effect. At the "OFF" state (stages 2 and 4), the charge transfer in the OCT-TENG is along with the movement of the slider, resulting in a relatively small current (Supplementary Figs. 9 and 10) and thus low energy output (Fig. 2c). This process is similar to that of the conventional sliding mode TENG. At room temperature and ambient environment, the total charge transfer of the OCT-TENG reaches ~3.7 μC per cycle (Fig. 2c), corresponding to 1.5 mC/m². The capacitor charging speed of the OCT-TENG is 2.7–3.0 times faster than the

control-TENG (Supplementary Figs. 11–13), which also verifies the opposite-charge-enhancement effect.

The transistor-like design allows the high current to be generated at the "ON" state, and the peak current is inversely proportional to the resistance in the circuit, as shown in Fig. 3a–e, j. Ideally, the internal resistance (output impedance) of the transistor-like TENG, $R_{in}$, is zero. But in real testing, a small $R_{in}$ can be detected from the circuit, which may come from the contact resistance. Fitting the measured current peak value with $I_{max} \propto (R_{in} + R)^{-1}$, where the $R_{in}$ is the internal resistance (output impedance) and $R$ is the load resistance, we get the $R_{in}$ of the TENG circuit of only 41 Ω (Supplementary Fig. 14). In contrast with the output current, the generated energy (average power) at the "ON" stage should be scaling with the $\frac{|Q_2 - Q_1|^2}{C_{TENG}}$, where $C_{TENG}$ is the capacitance of the TENG, which is unaffected from the load resistance (Fig. 3j). The U–Q plot at the "ON" state displays a jagged shape, rather than an ideal triangular, due to the split current curve generated by the air breakdown (Supplementary Fig. 15). Preventing the breakdown effect in future work can further enhance the performance of the OCT-TENG. Nevertheless, the output energy keeps the same order of magnitude at a large range of resistance, as shown in Fig. 3g, j.

At the "OFF" state, the power generation is along with the slider movement. Similar to the conventional TENG, the generated current at the "OFF" state is unaffected by the load resistance (when the resistance is relatively small) but highly affected by the sliding frequency, as shown in Fig. 3f, i, and Supplementary Fig. 16. Given the whole power generation of the

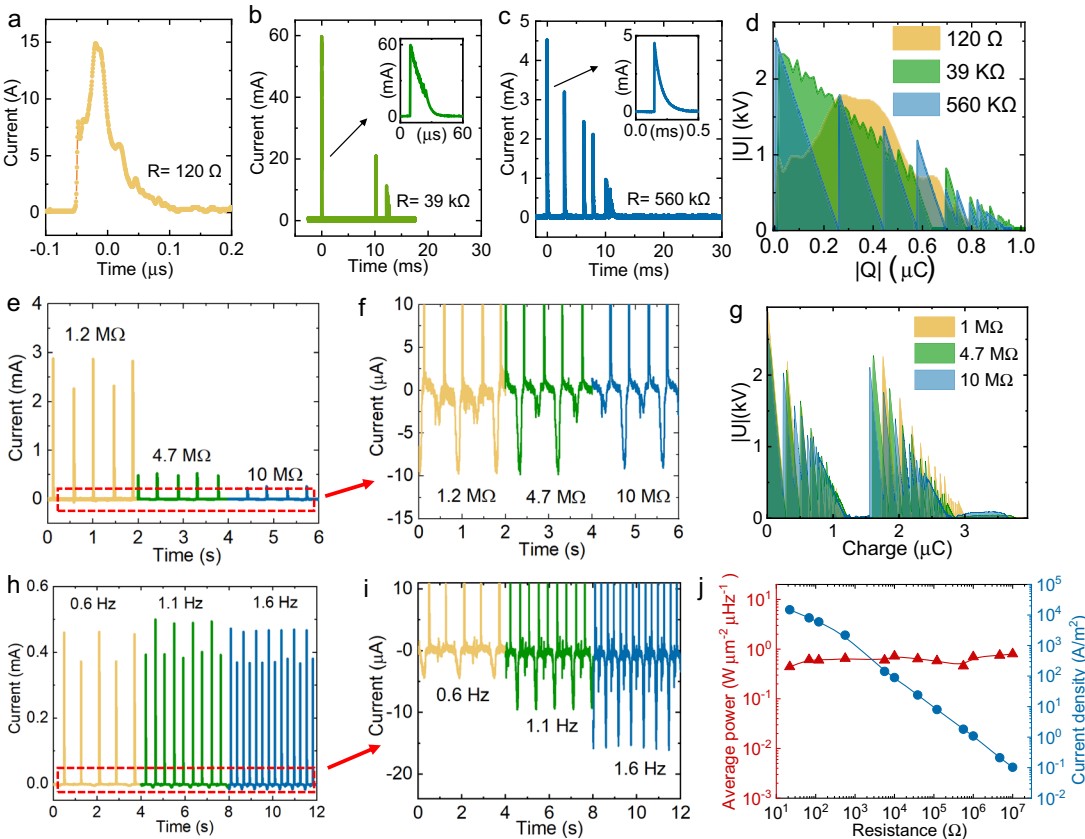

**Fig. 3 Evidence for the proposed mechanism.** The current output at stage 1 with the resistance of **a** 120 Ω, **b** 39 kΩ, and **c** 560 kΩ. **d** U–Q plot of OCT-TENG generated from stage 1 with the resistance of 120 Ω, 39 kΩ, and 560 kΩ. Output currents from the ON state (stages 1 and 3): **e** inversely proportional to the load resistance, and **h**, independent of the frequency. Output currents generated from the OFF state (stages 2 and 4): **f** independent of load resistance and **i** proportional to the frequency. **g** U–Q plot of the OCT-TENG with a load resistance of 1 MΩ, 4.7 MΩ, and 10 MΩ. **j** the average power density and peak current density depending on the load resistance.

OCT-TENG is dominated by the output from the "ON" stages, for a large range of load resistance from $22\,\Omega$ to $10\,M\Omega$, the average power densities keep roughly constant and all above $450\,mW\cdot m^{-2}\cdot Hz^{-1}$. The highest value of $790\,mW\cdot m^{-2}\cdot Hz^{-1}$ (with load resistance of $10\,M\Omega$) is higher than the previous record of $110\,mW\cdot m^{-2}\cdot Hz^{-1}$ [29]. Experiments based on varying parameters (including the effective area, the distance between $E_L$ and $E_R$, and the shape of the electrode on the stator) have also been performed, and all the results confirm our proposed mechanism (Supplementary Figs. 17–20).

**Direct observation of the opposite-charge-enhancement effect.** To demonstrate the opposite-charge-enhancement effect, we establish a "four-ports" methodology for directly detecting the opposite-charge-enhanced charge output of the OCT-TENG. As shown in Fig. 4a, we ground the four electrodes on the stator ($E_L, E_R, E_1$, and $E_2$) and monitor the charge output of $E_1$. In the case that $d_1, d_2 \gg d_3$, the charge transfer from $E_1$ to ground during stage 1 to stage 4 are $Q_2 - Q_1, -Q_2, 0$, and $Q_1$, respectively, as shown in Supplementary Fig. 21. Leveraging on this methodology, we clearly observed the charge accumulation as demonstrated in Fig. 4b, and the $Q_1$ and $Q_2$ can also be determined from the tested results, as shown in Fig. 4c. Despite the detection of the charge output, this "four-ports" method is also a quick and facile approach for evaluating different material systems for device optimization from the materials' perspective. We alter the tribo-material systems and test the value of $Q_1$ and $Q_2$ using the "four-ports" method, as shown in Fig. 4d and Supplementary Fig. 22. The high value of ($Q_2 - Q_1$) leads to the high current and power outputs of the OCT-TENG (Fig. 4e, Supplementary Figs. 23 and 24).

Due to the utility of ternary materials in our OCT-TENG, the $Q_1$ and $Q_2$ are affected by three tribo-materials. For FEP@FEP/PC

(refer to the tribo-material of the slider as FEP, and that of the stator as FEP and PC), the FEP surface on the stator also contains a certain amount of negative charges after rubbing with the identical material (FEP) on the slider. This is due to the balanced charge distribution after triboelectrification between the ternary material systems[36]. According to the electron-cloud-potential-well model[37], extra electrons occupying higher orbits are generated in the slider's FEP surface after its contact with PC on the stator. When the FEP on the slider contacts the FEP on the stator, these electrons occupying higher orbits tend to transfer to the empty orbits in the stator's FEP until the difference in potential-well depths between the surfaces is fully balanced. Therefore, the FEP on the stator also obtains negative charges, and this process is illustrated in Supplementary Fig. 25. Please note that all the charges on the materials' surfaces are generated from triboelectrification, and no other polarization process is employed in this work. So, there is no charge decay issue observed in our device. An OCT-TENG device has been stored in our lab in ambient condition for 6 months, without obvious degradation in the performance (Supplementary Fig. 26). But like all other TENG devices with physical contact, the long-term operation may also cause the wearing of the materials' surfaces in our OCT-TENG, as shown in Supplementary Figs. 27 and 28.

**Demonstration of the high-performance of the OCT-TENG.** The OCT-TENG can power low-power devices such as watches and thermometers (Supplementary Fig. 29 and Supplementary Video S5), and also high-power devices. Leveraging on the ultra-high-power output, we successfully light up one lamp rated 36 W, and five lamps rated 180 W using a small OCT-TENG with an effective area of 25 cm² (Fig. 5a–c, Supplementary Videos S1 and S2). Moreover, given the majority of power is generated at the "ON"

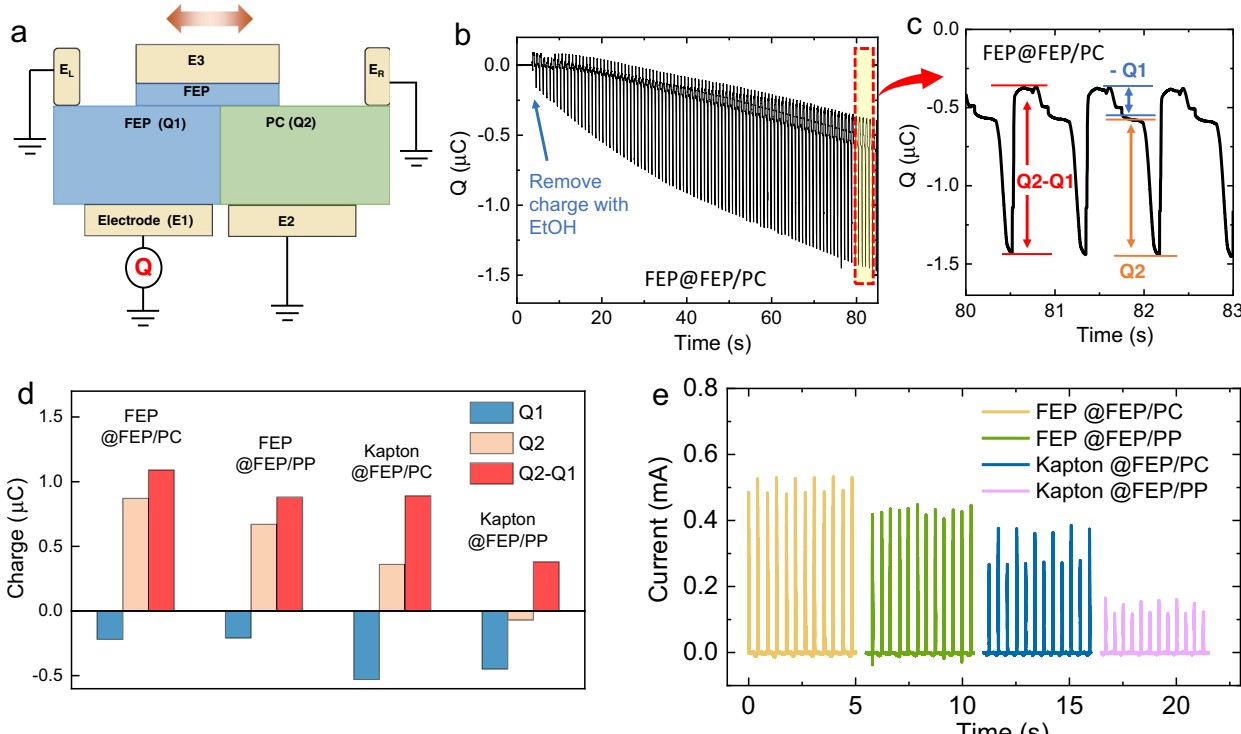

**Fig. 4 "Four-ports" methodology for evaluating the charge output. a** Schematic of the "four-ports" methodology for evaluating the charge output of the OCT-TENG. **b** Charge accumulation measured by the "four-ports" method. **c** charge on the FEP surface $Q_1$, charge on the PC surface $Q_2$, and the output charge of the OCT-TENG $Q_2 - Q_1$ measured by the "four-ports" method. **d** Measured $Q_1$, $Q_2$ and $Q_2 - Q_1$ with various material systems. (A@B/C refers to the tribo-surface of the slider as A and that of the stator as B and C). **e** Current output of the OCT-TENG (full circuit shown in Fig.1a) with various material systems (load resistance: 4.7 M$\Omega$).

state (stage 1 and stage 3) and the generated currents at the "ON" state are in the same direction, there is no need for a rectifier in the circuit. Such high-power output also allows us to power various electronic devices wirelessly. Using a simple power transmission circuit with two inductor coils (inductance: 50 µH, diameter: 52 mm), we achieve maximum power transmission efficiency of 75%, as shown in Fig. 5d and Supplementary Fig. 30. 825 mW-level LEDs and a vehicle bulb containing six commercial high-power LEDs (total power rated 30 W) are lighted using an OCT-TENG with the effective area of 25 cm$^2$ (Fig. 5e, f, Supplementary Videos 3 and 4). With the wirelessly powered vehicle bulb, objects at 0.9 meters away can be brightly illuminated, as shown in Fig. 5f.

## Discussion

In this work, we propose the OCT-TENG strategy for boosting the power output of the TENG by harnessing the combined effects from the opposite-charge synergy and the transistor-like device design. On the one hand, the amount of charge transfer has been enhanced by the opposite-charge-enhancement effect. On the other hand, utilizing the transistor-like structure, the power can be generated on the load devices with a close-to-zero impedance, leading to the ultra-high-power output of 12 MW/m$^2$ and average power density of 790 mW·m$^{-2}$·Hz$^{-1}$, far beyond previous records.

Benefiting from such high power, we directly light up 180 W commercial lamps and wirelessly power a 30 W vehicle bulb. Our OCT-TENG has demonstrated a device architecture framework for high-power TENGs and open up the avenues of using TENGs for powering electrical appliances and wirelessly powering the various electrical devices.

## Methods

**Fabrication of OCT-TENG**. The OCT-TENG contains a stator and a slider. In a material system of A@B/C, A is the tribo-surface on the slider; B and C are coplanar tribo-surfaces on the stator. The tribo-materials used in this work include FEP (fluorinated ethylene propylene), PC (polycarbonate), Kapton (polyimide), and PP (polypropylene). For the stator, the Kapton tape (100 µm) is used as substrate. On top of the Kapton tape, two 5 cm × 5 cm Cu tapes are fabricated as the electrode $E_1$ and $E_2$. ~100 µm film of Material A is placed on top of the $E_1$, and ~100 µm film of Material B is placed on top of the $E_2$. Conductive fabrics width of 0.5 cm are taped on two small acrylic flat cubes (2 cm × 2 cm × 0.3 cm) as the two floating electrodes $E_L$ and $E_R$. For the slider, a sponge is used as a substrate. A 6 cm × 5 cm sized conductive fabric is placed on the sponge as the electrode $E_3$. 5 cm × 5 cm film of Material C (10 µm) is placed on the conductive fabric as tribo-layer of the slider. The circuit is connected as the schematic shown in Fig. 1a with conductive wires (the conductive wires are wrapped with insulating coatings).

**Characterization**. The electric current lower than 5 mA is measured by a current preamplifier (SR570, Stanford Research System), and that for higher than 5 mA is calculated from the voltage across the load resistance measured by a digital storage

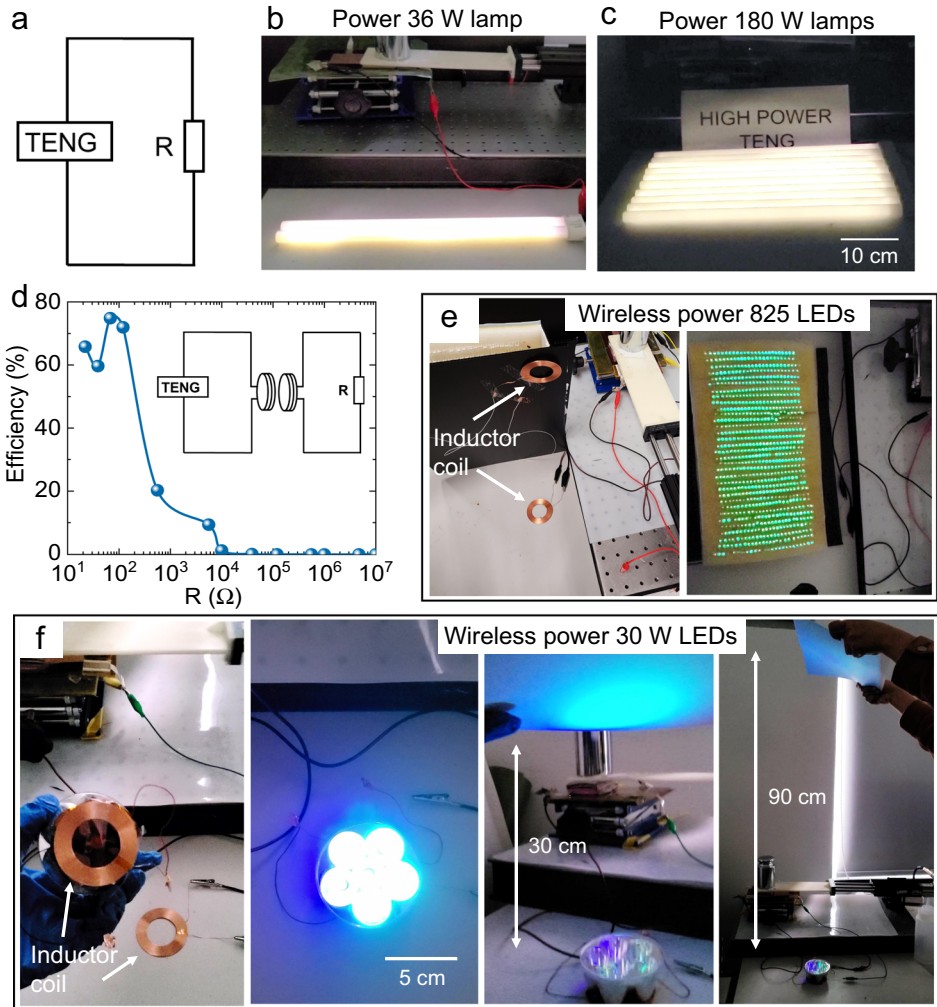

**Fig. 5 Demonstration of powering electrical devices by OCT-TENG. a** Circuit of directly powering the load devices using OCT-TENG. **b** A 36 W lamp and **c** 180 W lamps powered by the OCT-TENG. **d** wireless power transmission efficiency depending on the load resistance. Inset is the circuit of the wireless powering the load devices. **e** 825 LEDs being wirelessly powered by the OCT-TENG. **f** Vehicle bulb with 30 W LEDs powered by the OCT-TENG and illuminating objects 0.9 meters away.

oscilloscope (DSOX2014A, KEYSIGHT) equipped with a high-impedance (100 MΩ) probe. The output current of the control sample shown in Fig. 1b is also measured by the programmable electrometer (Keithley 6514), and the tested result is identical with that measured by the current preamplifier. The charge output of the "four-ports" method is measured by the programmable electrometer (Keithley 6514). The U–Q curves are based on measured current data with a known resistance $R$ as load. The test circuit is shown in Supplementary Fig. 31. The $U$ was calculated from $|U| = R|I(t)|$ and the $Q$ is calculated from $|Q| = \int_{t_0}^{t} |I| dt$, where $t_0$ is the time of the beginning of each cycle.

**Device performance demonstration**. The model of the high-power lamp we use for demonstrating the high-power output of the OCT-TENG is MASTER PL-L 36 W/ 840/4 P (PHILIPS) with rated power of 36 W, lamp length of 415 mm. The car lamp is assembled by six 5 W LEDs (in color of blue and green). The low-power LEDs with rated power of 60 mW are all in series connection. For the wireless powering, the inductance on the inductor coil is 50 μH, and the diameter of the coil is 52 mm. The power transmission efficiency is calculated as $\eta = E_T/E_0$, where $E_T$ is the energy wirelessly transferred on the load resistance via the power transmission circuit (inset of Fig. 5d), and $E_0$ is the energy delivered on the load resistance by directly connect the external load with the OCT-TENG (Fig. 5a). The energy was calculated as $E = \int I^2 R dt$, where the $I$ is current, $R$ is the load resistance and $t$ is time.

## Data availability
Source data are provided with this paper.

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

## Acknowledgements
This work was funded by HKSAR the Research Grants Council Early Career Scheme (Grant no. 24206919), HKSAR General Research Fund (Grant no. 14200120), and HKSAR Innovation and Technology Fund (Grant no. ITS/085/18), Research Grants Council of Hong Kong (No. C1006-20WF), Tencent Foundation through the XPLORER PRIZE.

## Author contributions
H.W. conceived the idea, performed the experiments, analyzed the data. H.W. drafted the manuscript with input from all authors. H.W., Y.Z., and Z.W. discussed the results. Y.Z. and Z.W. supervised this study. H.W., Y.Z., Z.W., and S.W. revised the manuscript. All authors reviewed the manuscript.

## Competing interests
The authors declare no competing interests.
