## [Peer Review File · Nature Communications]

Achieving Ultrahigh Instantaneous Power Density of 10 MW/m² by Leveraging the Opposite-Charge-Enhanced Transistor-Like Triboelectric Nanogenerator (OCT-TENG)REVIEWER COMMENTS

Reviewer #1 (Remarks to the Author):

TENG as a new energy technology that can convert various mechanical energy into electricity has attracted great attention. However, the TENG suffer from the high output impedance and low charge transfer. In this work, due to the opposite-charge-enhancement effect, the OCT-TENG can reach a pulsed power density over 10 MW/m², which set a record of the high-power output of TENG. And, the 180 W commercial lamps have been demonstrated to be lighted by the OCT-TENG. However, there are some problems need to be addressed in this work before publishing on Nature Communications.

1. The full name of PC needs to be noted in the main text.
2. It is hard to understand the charge transfer process in Figure 2a. We suggest revise this schematic diagram with charge symbols of different polarities to make it easy to understand for readers. The equivalent-circuit diagram of the TENGs is suggested to present.
3. Whether the U-Q curves in this work are directly measured. The test method should be described in the experimental section.
4. Because there is no triboelectrification process between FEP and FEP. How are the charges generated on the FEP films in the OCT-TENG? Is there a high voltage polarization process in the fabrication of the OCT-TENG. If so, whether the output of the OCT-TENG will decay over time. The stability and durability should be characterized.
5. The OCT-TENG has been demonstrated to power lamps of 180 W. What if using the OCT-TENG to power other electronic devices except lamp like watch, thermometers or other wearable electronic devices.

Reviewer #3 (Remarks to the Author):

This paper shows a novel discharging configuration for the free-standing-mode TENG. More importantly, the pulsed output power density reaches over 10 MW/m². The design is brilliant. I recommend it to be published after minor revisions:

1. Did the author measure the charging performance of the OCT-configuration? Please compare it with the original configuration to systematically exhibit the difference.
2. It would be nice for the authors to specify why there are so many output peaks in Fig. 2C.
3. The labels in Fig.3 d,g are faint, please revise.

Reviewer #4 (Remarks to the Author):

The authors report the ultrahigh power density of TENG achieved by leveraging the opposite-charge-enhancement. Overall, the approach is very interesting and the performance may be one of the best reported to date. However, I have few questions to answer before this work can be accepted in Nature Comm. Here are the comments.

1. The size of active layer is relatively large. How is the outputs of the device when the size is reduced gradually?
2. How is the output when the distance of EL and ER is reduced? The frequency can be kept constant if the size of stator is reduced proportionally. In addition, what is the output if the distance is reduced while keeping the same active layer area of 25 cm².
3. In Fig. 4d,e, do you observe the changes in negative peaks when differently pairing friction layers?
4. This work shows a very high output. However, because of the severe friction of the slider on the surface, the contact surface will be worn out quickly. Also, in the attached movie, I can observe the spark when "ON" state. How is the durability and stability of the device after long cyclic measurement? How long the highest output can be maintained? Please report the surface of the materials before and after long cyclic slide friction test?
5. In Fig 4d, A@B/C refers to the tribo-surface of the slider as A and that of the stator as B and C. Charge on the B surface is Q1, that on the C surface Q2. The Q1 on FEP (FEP@FEP/PC and FEP@FEP/PP) is identical for both FEP/ PC and FEP/PP, but Q1 on (Kapton @FEP/PC and Kapton@FEP/PP) is different. Please explain this.
6. In line '186', '195', and '196', there are typos.
In line 221, what is the Fig 4f?

Reviewer #1 (Remarks to the Author)

TENG as a new energy technology that can convert various mechanical energy into electricity has attracted great attention. However, the TENG suffer from the high output impedance and low charge transfer. In this work, due to the opposite-charge-enhancement effect, the OCT-TENG can reach a pulsed power density over 10 MW/m², which set a record of the high-power output of TENG. And, the 180 W commercial lamps have been demonstrated to be lighted by the OCT-TENG. However, there are some problems need to be addressed in this work before publishing on Nature Communications.

Response: We sincerely thank the reviewer for his/her compliment on our study and all the helpful comments. We now address the reviewer's concerns point-by-point as below.

1. The full name of PC needs to be noted in the main text.

Response: We thank the reviewer for this suggestion. The full name of PC is polycarbonates. We have added the full name of PC in the main text in manuscript according to the reviewer's suggestion. We also added the full name of all the tribo-materials we used in this work, including FEP (fluorinated ethylene propylene), PC (polycarbonate), Kapton (polyimide), and PP (polypropylene) into the Method section. The revisions are highlighted in yellow in the manuscript.

2. It is hard to understand the charge transfer process in Figure 2a. We suggest revise this schematic diagram with charge symbols of different polarities to make it easy to understand for readers. The equivalent-circuit diagram of the TENGs is suggested to present.

Response: We thank the reviewer for this helpful suggestion. We followed the reviewer's suggestion and revised the schematic in Fig. 2a by adding the charge symbols in the schematic. Please note the charge symbols used in Fig. 2a are for the material systems of FEP@FEP/PC. For some other material systems (such as Kapton@FEP/PP shown in Fig. 4d), the Q_1 and Q_2 may have different signs (refer to materials' polarizations). To make it clearer for the readers to understand the charge transfer processes in the OCT-TENG circuit, we also labeled the charge transfer path in each stage. The Fig. 2 has also been shown as **Fig. R1** below.

Fig. R1 (Revised Fig. 2) Working principle of the OCT-TENG. **a** Schematic diagram of the working principle of the OCT-TENG. The charge transfer path in each stage is labeled in red. **b** The generated current and **c** U-Q plot of the OCT-TENG during one cycle of operation (load resistance of 4.7 MΩ).

We added the equivalent-circuit as **Fig. R2** (also shown as Supplementary Fig. 9). **Fig. R2a** shows the full equivalent circuit. The TENG can be regarded as an equivalent voltage source and a varying capacitor connecting in series based on the standard theory. The V_1 and C_1 represent the equivalent voltage source and the capacitance between E_3 and E_1 , respectively. The V_2 and C_2 represent the equivalent voltage source and the capacitance between E_3 and E_2 , respectively. Equivalent circuits of the “ON” and “OFF” states are also shown in **Fig. R2b** and **2c**. At the “ON” state (stage I and stage III), the charge of $Q_2 - Q_1$ immediately transfer between the “Drain” and the “Source” (**Fig. R2b**). At the “OFF” state (stage II and stage IV), the charges transfer along with the movement of the slider (**Fig. R2c**).

Fig. R2 (Supplementary Fig. 9) (a) Equivalent circuit of OCT-TENG. (b) Equivalent circuit of OCT-TENG operated in Stage 1 and 3 (ON state). (c) Equivalent circuit of OCT-TENG operated in Stage 2 and 4 (OFF state).

The revised Fig. 2 and Supplementary Fig. 9, as well as the corresponding discussions are highlighted in yellow in the main manuscript and the Supplementary Information.

3. Whether the U-Q curves in this work are directly measured. The test method should be described in the experimental section.

Response: We thank a lot for the reviewer’s suggestion. The U-Q curves shown in this work are all based on measured data. The test method is as following. We measured the current output of the OCT-TENG with a known resistance (R) as load. The test circuit is shown in **Fig. R3** (Supplementary Fig. 31). The equipment we used for the current measurement is current preamplifier (SR570, Stanford Research Systems). We applied the denoise to the current curves for the low current during the “OFF” stages. In the U-Q curve, the U was calculated from $|U| = R|I(t)|$ and the Q was calculated from $|Q| = \int_{t_0}^t |I| dt$, where t_0 is the time of the beginning of each cycle (set $t_0 = 0$). Then the U-Q curve was plotted with $|Q|$ as x-axis and $|U|$ is y-axis.

Fig. R3 (Supplementary Fig. 31). The current measurement circuit for the U-Q curve.

We have followed the reviewer’s suggestion and added the description of the test method in the Method section, and the testing circuit as Supplementary Fig. 31 in the Supplementary Information. The revisions are highlighted in yellow.

4. Because there is no triboelectrification process between FEP and FEP. How are the charges generated on the FEP films in the OCT-TENG? Is there a high voltage polarization process in the fabrication of the OCT-TENG. If so, whether the output of the OCT-TENG will decay over time. The stability and durability should be characterized.

Response: We thank the reviewer for this comment. Below, we address the reviewer’s concerns point by point.

(1) We did not apply any “high voltage polarization process” in this work. All the charges on the materials’ surfaces were generated from the triboelectrification process. The reason that charges can be generated at FEP surfaces is that the rubbing between the FEP surface on the slider and the PC surface on the stator results in extra electrons occupying higher orbits in the slider’s FEP surface, and these electrons in the higher orbits tend to transfer to the empty orbits in the stator’s FEP and generate negative charges on it. We now revised the description in the manuscript to make this point clearer. We also revised the description in the Supplementary Information. The revision is as below:

“Due to the ternary material system in our OCT-TENG, the Q_1 and Q_2 are affected by the three tribo-materials. In the material system of FEP@FEP/PC (refer to the tribo-material of the slider as FEP, and that of the stator as FEP and PC), the FEP surface on the stator also contains a certain amount of negative charges after rubbing with the identical material (FEP) on the slider. This is due to the balanced charge distribution after triboelectrification of the ternary material systems³³. According to the electron-cloud-potential-well model³⁴, extra negative charges occupying higher orbits

are generated in the slider's FEP surface after it contact the PC on the stator. When the FEP on the slider contacts the FEP on the stator, these charges occupying higher orbits tend to transfer to the empty orbits in the stator's FEP until the difference in potential-well depths between the surfaces is fully balanced. Therefore, the FEP on the stator also obtains negative charges, and this process is illustrated in Supplementary Fig. 24.”

We also revised the Supplementary Fig. 24 and added more detailed description to make it clearer to the readers (Supplementary Fig. 24 is also shown as **Fig. R4**).

Fig. R4 (Supplementary Fig. 24). Explanation of the FEP surface on the stator gets negatively charged after contacting with FEP on the slide by electron cloud-potential well model. **a.** Electrons transfer from the PC surface to the FEP surface on the slider when the FEP on the slider contacts the PC surface **b.** Electrons transfer from the FEP surface on the slider to the FEP surface on the stator when the FEP on the slider contacts the FEP surface on the stator. **c.** After the contact of the three surfaces, positive charges are generated on the PC surface, and negative charges are generated on the FEP surfaces on both slider and the stator.

(2) Because we did not apply the “high voltage polarization process” in the fabrication

of the OCT-TENG, there is no charge decay problem in our device. The surface charge can be replenished during repeated triboelectrification. We re-tested our sample (in Jun. 2021) which was fabricated 6 months ago (in Dec. 2020), and the output does not change after 6 months stored in an ambient environment in our lab, as shown in **Fig. R5**.

Fig. R5 (Supplementary Fig. 26) Current outputs of the pristine OCT-TENG and the OCT-TENG stored in room environment for 6 months (load resistance: 4.7 M Ω).

(3) To better address the reviewer's concern on the durability of our OCT-TENG, we also performed a 10,000 cycles durability test with the OCT-TENG. Similar to most of the TENGs, the surface wear is the main issue for the OCT-TENG. Photographs of the samples before and after durability test has been shown in **Fig. R6a** and **R6b**. We also compared the materials' surfaces before and after durability test under microscope, as shown in **Fig. R6d**. For the optical microscope observation, to avoid the influence of the electrode background (such as Cu), we first peel the tribo-layers from the substrate and then observe it under microscope. The peeling process is shown in **Fig. R6c**. As shown in **Fig. R6d**, we found the wear on the PC surface was not obvious, while it's clear that the FEP surfaces have been worn. Considering that FEP is one of the most popular tribo-materials that is widely used in the TENG devices, overcoming its wearing issue is an interesting research topic in future. However, it's out of the scope of this study.

Fig R6 (Supplementary Fig. 27). Photographs of the OCT-TENG **a** before stability test and **b** after stability test. **c** Photograph of peeling the tribo-layer for the optical detection. **d** Observation of the tribo-surfaces (PC on the stator, FEP on the stator, and FEP on the slider) via optical microscope before and after stability test. Scale bar: 500 μm .

The results of the electric output during the stability test are shown in **Fig. R7**. Within 2500 cycles, there is no degradation of the output of the OCT-TENG. Current degradation has been observed from 3700 cycles. After 10000 cycles, both the current and voltage outputs show $\sim 20\%$ degradation. This was mainly caused by the wearing of the $10\ \mu\text{m}$ thin FEP film on the slider. From the surface observation of the microscope shown in **Fig. R6d** we can also observe that the FEP surface on the slider was obviously worn after the stability test. The electric output can fully recover after simply replacing the worn FEP film with a new FEP film on the slider, as shown in **Fig. R7 h-j**. Hence, maintenance that may be required after long-term operation is very quick and convenient.

Fig R7 (Supplementary Fig. 28) **a-h** The current output of the OCT-TENG with a load resistor of $4.7\text{ M}\Omega$ after various cycles' operation. The operation frequency is $\sim 1\text{ Hz}$. **i** Peak current depending on the cycle number. **j** Peak voltage depending on the cycle number.

We have added the above information into our main manuscript and the Supplementary Information (Supplementary Fig. 26-28), and the revision are highlighted in yellow.

5. The OCT-TENG has been demonstrated to power lamps of 180 W. What if using the OCT-TENG to power other electronic devices except lamp like watch, thermometers or other wearable electronic devices.

Response: We thank the reviewer for this comment. Compared to high-power lamps, the low-power devices such as watches or thermometers are much easier to be powered. Conventional TENGs can also power such low-power devices. Powering devices such as watches and thermometers needs a constant DC-power supply. So, suitable circuits

with capacitors, rectifiers, and/or transformers are needed according to the devices to be powered. Like many other TENGs, our OCT-TENG can also power watches and thermometers.

To better address the reviewer's concern, we performed additional experiments and demonstrated powering a sport watch and a thermometer using our OCT-TENG. The circuits and the pictures are shown in **Fig. R8**. The Video is shown in Supplementary Video S5.

Fig. R8 (Supplementary Fig. 29). **a** The circuit and **b** the photograph of powering a sport watch using the OCT-TENG (operation frequency: 0.8 Hz). **c** The circuit and **d** the photograph of powering a thermometer using the OCT-TENG (operation frequency: 0.8 Hz).

We have added this information and the figures and Videos into the main manuscript and the Supplementary Information (Supplementary Fig. 29). The revisions are highlighted in yellow.

Reviewer #3 (Remarks to the Author):

This paper shows a novel discharging configuration for the free-standing-mode TENG. More importantly, the pulsed output power density reaches over 10 MW/m². The design is brilliant. I recommend it to be published after minor revisions:

Response: We sincerely thank the reviewer for the favorable comments about our work and all the helpful suggestions. Below, we address in detail the specific comments raised by the reviewer.

1. Did the author measure the charging performance of the OCT-configuration? Please compare it with the original configuration to systematically exhibit the difference.

Response: We thank a lot for the reviewer's suggestion. We followed the reviewer's suggestion and performed the charging experiment of the OCT-TENG, and compared it with that of a control-TENG (the configuration shown in Fig.1b).

We first fabricated an OCT-TENG and a control-TENG, and measured their current outputs to make sure they both work well, as shown in **Fig. R9**. The operation frequency of the OCT-TENG and the control-TENG are both 1.2 Hz.

Then, we tested the charging performance of the OCT-TENG and the control-TENG. The capacitor charging circuits are shown in **Fig. R10**. In both circuits of OCT-TENG and the control TENG, a bridge rectifier is used to turn the AC into DC for the capacitor charging.

Fig. R11 shows the results of charging capacitors using the OCT-TENG and the control TENG. With an OCT-TENG, a 10 μF capacitor can be charged to 15V within 30 s. With 80 s charging time, the capacitors of 47 μF , 100 μF , and 220 μF can be charged to 10.30V, 4.15V, and 2.19 V, respectively (**Fig. R11a**). From the charging curves, we can observe the charge transfer at each stage, as shown in **Fig. R11b**. The charging performances of the control TENG with capacitors of 10 μF , 47 μF , 100 μF , and 220 μF are shown in **Fig. R11c** and **d**. The capacitor charging speed of the OCT-TENG is 2.7~3.0 times faster than the control TENG, as shown in **Fig. R11e-f**, confirming the opposite-charge enhancement effect of the OCT-TENG.

Fig. R9 (Supplementary Fig. 11) The generated current of the **a** OCT-TENG and **b** control TENG with a load resistance of $4.7\text{ M}\Omega$. The operation frequency of the OCT-TENG and control TENG are both 1.2 Hz .

Fig. R10 (Supplementary Fig. 12) The circuit of the charging the capacitors with **a**, OCT-TENG and **b**, control TENG.

Fig. R11 (Supplementary Fig. 13) The charging curves of 10 μF (black), 47 μF (red), 100 μF (green), and 220 μF (blue) capacitors with (a, b) OCT-TENG and (c, d) control TENG. The operation frequency of OCT-TENG and control TENG are both 1.2 Hz (shown in Fig. R9). The charging curves of (e) 10 μF , (f) 10 μF , and (g) 10 μF capacitors with OCT-TENG and the control TENG.

We thank the reviewer for suggesting these experiments and we are happy to see that based on these experimental results, we further validated and better illustrated the opposite-charge enhancement effect of the OCT-TENG proposed in this study. We have added a short discussion in our main manuscript and the information into Supporting Information. The revisions are highlighted in yellow.

2. It would be nice for the authors to specify why there are so many output peaks in Fig. 2C.

Response: At the “ON” stage, a large quantity of charges ($Q_2 - Q_1$) immediately transfers between the E_3 on the slider and the E_R/E_L on the two sides of the stator. This causes the air breakdown when the E_3 moves close to the E_R or E_L . A portion of the unbalanced charges transfer between E_3 and E_R/E_L during air breakdown until the unbalanced charges are insufficient to keep the air breakdown voltage. One current peak is therefore generated. After that, the slider keeps moving towards the E_R or E_L , and at a certain position the air breakdown happens again due to the reduced distance between the E_3 and the E_R/E_L , another portion of the charges transfer during the air breakdown. As a result, a current peak is therefore generated again. After several times of air breakdown, the E_3 touches the E_R/E_L , and then all the unbalanced charges transferred between E_3 and E_R/E_L . Therefore, the air breakdown effect leads to the current curve split and appears several peaks. We added a short discussion in the main text.

The revision has been highlighted in yellow.

3. The labels in Fig.3 d,g are faint, please revise.

Response: We appreciate the reviewer for the careful check of our manuscript. We have revised the Fig. 3.

Reviewer #4 (Remarks to the Author)

The authors report the ultrahigh power density of TENG achieved by leveraging the opposite-charge-enhancement. Overall, the approach is very interesting and the performance may be one the best reported to date. However, I have few questions to answer before this work can be accepted in Nature Comm. Here are the comments.

Response: We sincerely thank the reviewer for his/her compliment on our work and all the helpful suggestions. Below, we address the specific comments raised by the reviewer point by point.

1. The size of active layer is relatively large. How is the outputs of the device when the size is reduced gradually?

Response: We thank the reviewer for this comment. In theory, according to our model, with the identical material system the charge output should be proportional to the effective area of the sample. Because the charge transfer at the “ON” stage is $Q_2 - Q_1 = (\sigma_2 - \sigma_1)A$, where A is the effective area (refer to one Cu electrode area on the stator in this work), the σ_1 and σ_2 are the charge densities corresponding to Q_1 and Q_2 . With the identical material system, the generated charge densities on the material’s surface should be roughly the same.

To better address the reviewer’s concern, we performed experiments on the devices with reduced effective areas, and the results is shown in **Fig. R12-13**. **Fig. R12a** demonstrates the photos of the three OCT-TENG devices fabricated with effective areas of 5 cm × 5 cm (Device #1), 4 cm × 4 cm (Device #2), and 3 cm × 3 cm (Device #3). In this comparison experiment, we used the same electrode of E_L and E_R for the Device #1, Device #2 and Device #3. The OCT-TENG with E_L and E_R is shown in **Fig. R12b**.

We first tested the charge output using the “four-ports” methodology which has been demonstrated in Fig. 4 in our manuscript. As shown in **Fig. R12c-e**, $Q_2 - Q_1$ of Device #1, Device #2, and Device #3 detected using the “four-ports” methodology are 1.12 μC , 0.67 μC , and 0.33 μC respectively. It is clear that the charge output of $Q_2 - Q_1$ is scaled with the effective area.

Fig. R12 (Supplementary Fig. 17) (a) Photograph of the OCT-TENG (without “Gate” electrodes of E_L and E_R) with various effective areas of 5 cm × 5 cm (Device #1), 4 cm × 4 cm (Device #2), and 3 cm × 3 cm (Device #3). (b) Photograph of the OCT-TENG with electrodes of E_L and E_R . Charge output of (c) Device #1, (d) Device #2, and (e) Device #3 detected via the “four-ports” methodology. Current outputs of (f) Device #1, (g) Device #2, and (h) Device #3 with load resistance of 4.7 MΩ. The generated current of the OCT-TENG at the “ON” state of (i) Device #1, (j) Device #2, and (k) Device #3 with load resistance of 4.7 MΩ. The values of $|Q_2 - Q_1|$ of Device #1, Device #2, and

Device #3 calculated from the current integration are shown in (i – k) in red.

Fig. R13 (Supplementary Fig. 18) The charge transfer at ON state (stage 1 or 3) depending on the effective area of OCT-TENG.

We then compared the current outputs (load resistance: 4.7 MΩ), and the results are shown in **Fig.R12 f-k**. The current of the three devices are roughly similar, and the peak values are slightly decreased with the area decreasing. At the “ON” stage, charges of $Q_2 - Q_1 = (\sigma_2 - \sigma_1)A$ immediately transfer between the “Source” and the “Drain”. The current peak $I_p = \frac{U}{R} \propto \frac{(\sigma_2 - \sigma_1)}{cR}$, where the $(\sigma_2 - \sigma_1)$ are the unbalanced charges, and c is the dielectric capacitance per area, the R is the load resistance. As a result, the current peak value is in theory identical when varying the area due to the similar $\sigma_2 - \sigma_1$. However, practically, when the area is small, the total transferred charges at the “ON” stage ($Q_2 - Q_1$) is relatively small and is easier to be affected by factors such as the testing instruments, the circuit, and the environment.

Despite the similar peak currents, the charge transfer at the “ON” stages integrated from the current show that the $Q_2 - Q_1$ of the three devices decrease with the active area decreasing. The value of $Q_2 - Q_1$ integrated from the current at the “ON” state are 1.09 μC, 0.64 μC and 0.42 μC for Device #1, Device #2 and Device #3, respectively. These values are consistent with the results tested using the “four-ports” methodology.

Thus, the charge output decreases linearly with the size of the device. (**Fig. R13**)

We thank the reviewer for suggesting these experiments, and the results further confirmed our proposed working mechanism of the OCT-TENG. We now have added the above information into the Supplementary Information (Supplementary Fig. 17-18) and highlighted them in yellow.

2. How is the output when the distance of EL and ER is reduced? The frequency can be kept constant if the size of stator is reduced proportionally. In addition, what is the output if the distance is reduced while keeping the same active layer area of 25 cm².

Response: We followed the reviewer's suggestion and performed the experiments by varying the distance of the E_L and E_R on the same OCT-TENG device with effective area of 25 cm². The photograph of the experiments and the results are shown in **Fig. R14**.

As shown in **Fig. R14 a-c**, we set the distance d of the E_L and E_R as 11 cm, 9 cm, and 7 cm, respectively. **Fig. R14d-e** show the generated current when varying the d . Although the area of the device is kept constant, the effective area is decreased due to the decrease of d . As a result, the transferred charge $Q_2 - Q_1$ at the "ON" stage decreased with the decrease of the d , as shown in **Fig. R14 d-i**.

Fig R14 (Supplementary Fig. 19). The photographs of the OCT-TENG with the distance of E_L and E_R of (a) 11 cm, (b) 9 cm, and (c) 7 cm. The generated current of the OCT-TENG with the distance of E_L and E_R of (d) 11 cm, (e) 9 cm, and (f) 7 cm. (load resistance is $4.7 \text{ M}\Omega$). The charge transfer, $|Q_2 - Q_1|$, of the OCT-TENG at the “ON” state calculated from the current integration with the distance of E_L and E_R of (g) 11 cm, (h) 9 cm, and (i) 7 cm. (load resistance is $4.7 \text{ M}\Omega$).

To better address the reviewer’s concern, we further performed experiments and compared the outputs of two OCT-TENG with identical effective area but different electrode shapes, also corresponding to different distance between E_L and E_R . The photograph of the OCT-TENG with effective area of $3.1 \text{ cm} \times 8 \text{ cm} \approx 25 \text{ cm}^2$ is shown in **Fig. R15a**. As a comparison, the standard OCT-TENG with area of $5 \text{ cm} \times 5 \text{ cm} = 25 \text{ cm}^2$ is shown in **Fig. R15b**. As shown in **Fig. R15c-d**, the current outputs (load resistance of $4.7 \text{ M}\Omega$) of these two devices are very similar. The $Q_2 - Q_1$ calculated from the integral of the current at “ON” state are also similar, as shown in **Fig. 15 e-f**. We also performed the “four-ports” methodology to further detect the quantity of the charge transfer, and we found the measured $Q_2 - Q_1$ for these two devices are both $1.12 \text{ }\mu\text{C}$, as shown in **Fig. 15g-h**. So, although the shapes of these two devices are different, as long as their effective areas are identical, the outputs will be almost

identical.

Fig. R15 (Supplementary Fig. 20). (a) Photograph of the OCT-TENG with effective area of $3.1 \text{ cm} \times 8 \text{ cm} \approx 25 \text{ cm}^2$. (b) Photograph of the stators of the OCT-TENGs with effective area of $3.1 \text{ cm} \times 8 \text{ cm}$ and $5 \text{ cm} \times 5 \text{ cm}$. The generated current of the OCT-TENGs with the area of (c) $3.1 \text{ cm} \times 8 \text{ cm}$ and (d) $5 \text{ cm} \times 5 \text{ cm}$. The charge transfer, $|Q_2 - Q_1|$, at the “ON” state calculated from the current integration of OCT-

TENGs with the area of (e) $3.1 \text{ cm} \times 8 \text{ cm}$ and (f) $5 \text{ cm} \times 5 \text{ cm}$. Charge output of OCT-TNEGs with the area of (g) $3.1 \text{ cm} \times 8 \text{ cm}$ and (h) $5 \text{ cm} \times 5 \text{ cm}$.

We sincerely thank the reviewer for suggesting these experiments that help better demonstrate and confirm the working mechanism of the OCT-TENG. We added the above experimental results and discussion into the Supplementary Information, and short discussion in the main manuscript as well. The revisions are highlighted in yellow.

3. In Fig. 4d,e, do you observe the changes in negative peaks when differently pairing friction layers?

Response: Yes, we observed the differences in the negative peaks when differently pairing friction layers. We show the negative current peaks generated from samples with various tribo-material pairs in Fig. R16. It is obvious that the peak value of the negative currents of the samples is scaled with the maximum value of $-Q_1$ and Q_2 , which is consistency with the proposed mechanism.

Fig. R16 (Supplementary Fig. 23). The negative current peaks generated from OCT-TENGs with various tribo-material pairs.

We have added this figure into Supplementary Information and highlighted the revision in yellow.

4. This work shows a very high output. However, because of the severe friction of the slider on the surface, the contact surface will be worn out quickly. Also, in the attached movie, I can observe the spark when "ON" state. How is the durability and stability of the device after long cyclic measurement? How long the highest output can be maintained? Please report the surface of the materials before and after long cyclic

slide friction test?

Response: We thank the reviewer for this comment. Indeed, because the charges are generated from the triboelectrification, wear is an issue for almost all the TENGs, including our OCT-TENG. Some reports have proposed strategies to circumvent this problem, such as non-contact mode operation, adding a lubricating layer, or using liquid as the tribo-material.

The spark appeared at the "ON" state between the electrode on the slider (E_3) and the "Gate" electrode (E_L and E_R) is due to the air breakdown effect. The air breakdown is caused by the large amount of the charge ($Q_2 - Q_1$) transfer at a short time when the slider E_3 is approaching the gate electrode (E_L or E_R). The air breakdown between two electrodes hardly causes any wearing issue. So, like all other TENGs with physical contact, the main durability limitation in our OCT-TENG is also the wearing of the tribo-materials.

We then followed the reviewer's advice and performed the stability test on the OCT-TENG. Photograph of the samples before and after stability test has been shown in *Fig. R17a* and *R17b*. We also compared the materials' surfaces before and after the stability test under microscope, as shown in *Fig. R17d*. For the optical microscope observation, to avoid the influence of the electrode background (such as Cu), we first peel the tribo-layers from the substrate and then observe it under microscope. The peeling process is shown in *Fig. R17c*. As shown in *Fig. R17d*, we found the wear on the PC surface was not obvious, while it's clear that the FEP surfaces have been worn. Considering that FEP is one of the most popular tribo-materials that is widely used in the TENG devices, addressing its wearing issue is an interesting topic. However, it's out of the scope of this study.

Fig R17 (Supplementary Fig. 27). Photographs of the OCT-TENG **a** before and **b** after stability test. **c** Photograph of peeling the tribo-layer for the optical detection. **d** Observation of the tribo-surfaces (PC on the stator, FEP on the stator, and FEP on the slider) via optical microscope before and after stability test. Scale bar: 500 μm .

The results of the electric output during the stability test are shown in **Fig. R18**. Within 2500 cycles, there is no degradation of the output of the OCT-TENG. Current degradation has been observed from 3700 cycles. After 10000 cycles, the current and voltage outputs show $\sim 20\%$ degradation. This was mainly caused by the wearing of the 10 μm thin FEP film on the slider. From the surface observation of the microscope shown in **Fig. R17d** we can also observe that the FEP surface on the slider was obviously worn after the stability. The electric output can fully recover after simply replacing the worn FEP film with a new FEP film on the slider, as shown in **Fig. R18 h-j**. Hence, the maintenance that may be required after long-term operation will be very quick and convenient.

Fig R18 (Supplementary Fig. 28). **a-h** The current output of the OCT-TENG with a load resistor of $4.7 \text{ M}\Omega$ after various cycles' operation. The operation frequency is $\sim 1 \text{ Hz}$. **i** Peak current depending on the cycle number. **j** Peak voltage depending on the cycle number.

We have added the above information into our main manuscript and the Supplementary Information, and the revision are highlighted in yellow.

5. In Fig 4d, A@B/C refers to the tribo-surface of the slider as A and that of the stator as B and C. Charge on the B surface is Q_1 , that on the C surface Q_2 . The Q_1 on FEP@FEP/PC and FEP@FEP/PP is identical for both FEP/PC and FEP/PP, but Q_1 on (Kapton @FEP/PC and Kapton@FEP/PP) is different. Please explain this.

Response: We thank the reviewer for raising this question. In the OCT-TENG, the charges are generated from the triboelectrification between the tribo-materials. According to the electron-cloud-potential-well model (Xu C, Zi Y, Wang A C, et al. *Advanced*

Materials, 2018, 30(15): 1706790.), the charges occupying higher orbits in one material tend to transfer to the empty orbits in the other material until the difference in potential-well depths between the surfaces is fully balanced. In our OCT-TENG, three materials are involved in the tribo-process, including the material A on the stator, the material B, and C on the stator. So, the Q_1 depends on the potential balance of three tribo-materials involved in the system. It means Q_1 depends on not only the FEP on the stator, but also the other tribo-material on the stator and the tribo-material on the slider.

We illustrate the tribo-charge generation process in **Fig. R19**. In material system of A@B/C, when A on the slider contacts the C on the stator, electrons transfer between C to A and generate a certain number of tribo-charges on the surfaces of A and C. Then when A contacts B, electrons transfer between A and B and thus tribo-charges are generated on the B's surface. Consequently, the Q_1 generated on the FEP surface (Material B) on the stator not only depends on the materials on the slider (Material A), but also is affected by the triboelectrification process between the Material A and Material C on the stator, as reflected by Q_2 .

From Figure 4d, we can see that after the tribo-process, the Q_2 as the charge generated in material C of FEP@FEP/PC and FEP@FEP/PP are both positive and their difference in value are relatively small, and thus the impact of different material C on the potential balance in the ternary material system is little. So that the Q_1 for FEP@FEP/PC and FEP@FEP/PP are also similar, and the values are 0.22 μC and 0.21 μC . However, for the material systems of Kapton @FEP/PC and Kapton@FEP/PP, the values of Q_2 are largely different. The Q_2 for Kapton @FEP/PC is positive while the PP surface of Kapton @FEP/PP is slightly negatively charged (Q_2 is negative). This reflects the triboelectrification of Kapton and PC is very different from that of Kapton and PP, which may be due to distinct mechanisms that deserves further investigations in the future. The different triboelectric effect of the Kapton in contact with PP and PC affect the tribo-charges generated on the FEP surface on the stator (Q_1). As a result, the Q_1 shows much difference in the systems of Kapton@FEP/PC and Kapton@FEP/PP.

Fig. R19. Illustration of the tribo-charge generated at materials' surfaces at a system of A@B/C.

6. In line '186', '195', and '196', there are typos. In line 221, what is the Fig 4f?

Response: We appreciate the reviewer for the careful check of our manuscript. We have carefully checked our manuscript and revised the typos in the manuscript.

REVIEWERS' COMMENTS

Reviewer #1 (Remarks to the Author):

The authors clearly responded to the reviewer's comments. And, the modified manuscript is well organized. I recommend publication in Nature Communications for the paper in its present form.

Reviewer #3 (Remarks to the Author):

The authors have fully addressed my questions. Here I have a few more questions:

1. The power output reported here is not an average power output but an instantaneous output, which was achieved by using the accumulative role of the transistor like TENG. Such idea was previously reported in:

"Pulsed Nanogenerator with Huge Instantaneous Output Power Density", ACS Nano, 7 (2013) 7383-7391.

Please quote this original paper.

2. In the title and in the text, the $10\text{MW}/\text{m}^2$ quoted is the instantaneous output power. To avoid a confusion of the reader it being the average power. The authors need to use the term of "instantaneous output power density of $10\text{MW}/\text{m}^2$ " in the title and in the text. This is good for the authors and the readers to avoid confusion and misunderstanding.

Reviewer #4 (Remarks to the Author):

The authors answered to the most of questions raised by the reviewer.

Reviewer #1 (Remarks to the Author)

The authors clearly responded to the reviewer's comments. And, the modified manuscript is well organized. I recommend publication in Nature Communications for the paper in its present form.

Response: We sincerely thank the reviewer for his/her compliments and the recommendation of our paper to be published in Nature Communications.

Reviewer #3 (Remarks to the Author)

The authors have fully addressed my questions. Here I have a few more questions:

1. The power output reported here is not an average power output but an instantaneous output, which was achieved by using the accumulative role of the transistor like TENG. Such idea was previously reported in:

“Pulsed Nanogenerator with Huge Instantaneous Output Power Density”, ACS Nano, 7 (2013) 7383-7391.

Please quote this original paper.

Response: We thanks a lot to the reviewer for pointing this out. We have in fact already included the results of this paper in Fig. 1e, but we are very sorry for missing this paper in the reference by mistake. We now well quoted this paper and added the information in the References. We also added the references related to Fig. 1e in the manuscript and the captions of Fig. 1.

2. In the title and in the text, the $10\text{MW}/\text{m}^{**2}$ quoted is the instantons output power. To avoid a confusion of the reader it being the average power. The authors need to use the term of "instantons output power density of $10\text{MW}/\text{m}^{**2}$ " in the title and in the text. This is good for the authors and the readers to avoid confusion and misunderstanding.

Response: We thank the reviewer for this helpful suggestion. We followed the reviewer's suggestion and adjusted the title of our paper as “Achieving Ultrahigh Instantaneous Power Density of $10\text{MW}/\text{m}^2$ by Leveraging the Opposite-Charge-Enhanced Transistor-Like Triboelectric Nanogenerator (OCT-TENG)”.

In fact, the phrase in our original manuscript of “pulsed power density” technically refers to “instantaneous power density”. We also used the “average power density” when we discuss the average power density of our device and compare it with previous reports. But for avoiding misunderstanding, we followed the reviewer’s suggestion and rephrase the “pulsed power density” as “instantaneous power density.”

Reviewer #4 (Remarks to the Author)

The authors answered to the most of questions raised by the reviewer.